Variation in female reproductive tract morphology across the reproductive cycle in the zebra finch

http://orcid.org/0000-0003-1688-6662 Hurley Laura L. 1 Laura.Hurley@mq.edu.au
Crino Ondi L. 2
http://orcid.org/0000-0001-9747-041X Rowe Melissah 3
http://orcid.org/0000-0001-7612-4999 Griffith Simon C. 1
1 Department of Biological Sciences, Macquarie University , Sydney, NSW , Australia
2 School of Life and Environmental Sciences, Deakin University , Burwood, VIC , Australia
3 Department of Animal Ecology, Netherlands Institute of Ecology , Wageningen , Netherlands
Harrison Xavier
Electronic publication date: 2020 Nov 11
Publication date: 2020
Volume: 8
Electronic Location ID: e10195
Received 2020 Jun 17; Accepted 2020 Sep 24
Copyright: © 2020 Hurley et al.
Copyright year: 2020
Copyright holder: Hurley et al.
License: This is an open access article distributed under the terms of the Creative Commons Attribution License, which permits unrestricted use, distribution, reproduction and adaptation in any medium and for any purpose provided that it is properly attributed. For attribution, the original author(s), title, publication source (PeerJ) and either DOI or URL of the article must be cited.
License URL: https://creativecommons.org/licenses/by/4.0/

Keywords: Finch, Opportunistic breeder, Ovarian development, Oviduct, Reproduction

Funding: Australian Research Council DP130100417 Research Council of Norway 230434 This work was supported by Australian Research Council (No. DP130100417 to Simon C. Griffith, Katerine L. Buchanan, and Melissah Rowe). Melissah Rowe was supported by a Young Research Talent grant from the Research Council of Norway (No. 230434). Simon C. Griffith is an Academic Editor for PeerJ. The funders had no role in study design, data collection and analysis, decision to publish, or preparation of the manuscript.

==============================
Background

In seasonally breeding birds, the reproductive tract undergoes a dramatic circannual cycle of recrudescence and regression, with oviduct size increasing 5–220 fold from the non-breeding to the breeding state. Opportunistically breeding birds can produce multiple clutches sequentially across an extended period in response primarily to environmental rather than seasonal cues. In the zebra finch, it has been shown that there is a significant reduction in gonadal morphology in non-breeding females. However, the scale of recrudescence and regression of reproductive tissue within a single breeding cycle is unknown and yet important to understand the cost of breeding, and the physiological readiness to breed in such flexible breeders.

Methods

We examined the reproductive tissue of breeding female zebra finches at six stages in the nesting cycle from pre-breeding to fledging offspring. We quantified the wet mass of the oviduct, the volume of the largest pre-ovulatory follicle, and the total number of pre-ovulatory follicles present on the ovary.

Results

Measures of the female reproductive tract were highest during nesting and laying stages and declined significantly in the later stages of the breeding cycle. Importantly, we found that the mass of reproductive tissue changes as much across a single reproductive event as that previously characterized between birds categorized as breeding and non-breeding. However, the regression of the ovary is less dramatic than that seen in seasonal breeders. This could reflect low-level maintenance of reproductive tissues in opportunistic breeders, but needs to be confirmed in wild non-breeding birds.

Introduction

The avian reproductive system is typically viewed as a dynamic structure that exhibits dramatic changes in size and development across seasons (Williams, 2012). For example, in seasonally breeding birds, the reproductive tract undergoes a circannual cycle of regression and recrudescence (Johnson & Woods, 2007; Keck, 1934; Wingfield & Farner, 1993). In passerines, measurements taken during non-breeding and breeding periods show that oviductal wet mass increases by up to 220-fold during the breeding season, while the linear dimensions of the largest follicle have been shown to increase by up to 60-fold (see Table S1). The reproductive cycle in seasonally breeding birds is primarily regulated by day length and, to a lesser extent, is finely tuned by environmental conditions such as rainfall, temperature, and access to nesting sites (Dawson et al., 2001; Wingfield & Farner, 1993; Wingfield et al., 1992). In temperate environments with distinct seasons, day length is a reliable indicator of food availability. Therefore, seasonal environments are thought to exert strong selective pressure on reproductive physiology that allows birds to time reproduction during peak seasonal food availability.

Table 1 Pairwise comparison between breeding stages.

Post hoc comparison between six reproductive time points: pre-breeding (pre), nesting, laying, incubation, post-hatch, and fledging for both oviduct mass and the volume of the largest pre-ovulatory follicle present on the ovary in female zebra finches. Comparisons tested using emmeans with P-value adjustment for multiple comparisons using Tukey HSD. Significant P-value denoted in bold.

	Oviduct mass	Follicle volume	
Contrasts	est.	SE	df	t ratio	P-value	est.	SE	df	t ratio	P-value	
Pre-Nesting	−0.22	0.30	27	−0.75	0.97	−0.70	0.76	29	−0.92	0.94	
Pre-Laying	−0.79	0.31	27	−2.57	0.14	0.14	0.78	29	0.19	1.00	
Pre-Incubation	1.22	0.30	27	4.13	0.004	2.24	0.76	29	2.94	0.06	
Pre-PostHatch	1.79	0.32	27	5.57	<0.001	2.25	0.78	29	2.88	0.07	
Pre-Fledging	1.68	0.32	27	5.31	<0.001	2.44	0.79	29	3.10	0.05	
Nesting-Laying	−0.57	0.29	27	−1.97	0.3825	0.85	0.78	29	1.08	0.89	
Nesting-Incubation	1.44	0.28	27	5.14	<0.001	2.94	0.76	29	3.86	0.007	
Nesting-PostHatch	2.01	0.30	27	6.65	<0.001	2.95	0.79	29	3.74	0.01	
Nesting-Fledging	1.90	0.30	27	6.44	<0.001	3.14	0.80	29	3.94	0.006	
Laying-Incubation	2.01	0.29	27	6.97	<0.001	2.09	0.78	29	2.68	0.11	
Laying-PostHatch	2.58	0.29	27	8.77	<0.001	2.11	0.76	29	2.76	0.09	
Laying-Fledging	2.47	0.28	27	8.79	<0.001	2.30	0.77	29	3.00	0.06	
Incubation-PostHatch	0.57	0.30	27	1.88	0.43	0.01	0.79	29	0.02	1.00	
Incubation-Fledging	0.46	0.30	27	1.57	0.63	0.20	0.80	29	0.25	1.00	
PostHatch-Fledging	−0.11	0.30	27	−0.36	1.00	0.19	0.76	29	0.25	1.00	

However, not all bird species breed during discreet or predictable periods of time. Some species exhibit flexible breeding periods or breed opportunistically when conditions become favorable. In opportunistically breeding bird species, reproduction is timed primarily through non-photic cues such as food availability, temperature, and rainfall, and displays a dampened seasonal peak (Englert Duursma, Gallagher & Griffith, 2019; Hahn et al., 1997; Hahn et al., 2008; Hau, Perfito & Moore, 2008). In contrast to seasonally breeding birds, opportunistically breeding species are hypothesized to maintain some degree of gonadal development year-round in order to rapidly initiate reproduction when ecological conditions become optimal (Wingfield, 2008).

Compared to the well-established literature on seasonally breeding birds, relatively few studies have investigated variation in reproductive morphology in opportunistically breeding birds. The studies that have been conducted suggest that there is annual variation in gonad development, but full regression is not always evident (Hahn, 1998; Perfito et al., 2007). This supports the hypothesis that the there is some maintenance of gonads potentially allowing opportunistic breeders to rapidly exploit reproductive opportunities. For example, in wild zebra finches (Taeniopygia guttata), Perfito et al. (2007) reported a mean follicle size (for the largest follicle in each female) of 3 mm3 and 44 mm3 (early and late within a breeding season, respectively) and a mean follicle size (of the largest follicle) of around 5 mm3 in a non-breeding population (all values estimated from Fig. 2 in Perfito et al. (2007)). These data suggest that the smallest values found in breeding birds are equivalent to those in non-breeding populations. However, an important caveat is that the exact breeding status of the females studied by Perfito et al. (2007) was uncertain. In a laboratory study, Prior, Heimovics & Soma (2013) used a water deprivation treatment to significantly inhibit reproduction in females. Specifically, Prior, Heimovics & Soma (2013) reported that the largest follicle diameter in each female remained 1.8 mm on average. These data suggest follicle volume in water restricted non-breeding females averaged 3.05 mm3 (using V = 4/3πa3 where V = volume and a = radius of follicle), making them somewhat smaller than those reported by Perfito et al. (2007) in a non-breeding wild population in an unpredictable environment, but equivalent to late-breeding in a predictable environment. Importantly, however, these values are greater than the minimum size of the fully regressed follicles reported in seasonal breeders (see Table S1).

The significant shift in follicle volume and oviduct mass in water restricted zebra finches, did not coincide with a change in estradiol levels (Prior, Heimovics & Soma, 2013). Further, in wild zebra finches detectable estradiol was found in one of nine non-breeding birds in an unpredictable environment, but in none of the five birds tested when breeding in the predictable environment (Perfito et al., 2007). Both of these findings are surprising given that in birds, estradiol aids in regulating reproduction by stimulating the production of yolk precursors from the liver (Wallace, 1985), and stimulating oviduct development (Pollock & Orosz, 2002; Williams, 1999). It is possible that changes in estradiol will only be seen during nesting and egg laying when it is stimulating yolk formation and oviduct growth and development (Pollock & Orosz, 2002; Wallace, 1985; Williams, 1999). However, studies conducted thus far have not looked at estradiol production in zebra finches at this fine temporal resolution.

Although past studies have established that opportunistically breeding birds broadly maintain a degree of gonadal development when not breeding, no study to date has examined fine scale variation in gonad development within and between reproductive cycles. The average follicle sizes of the non-breeding zebra finches (Perfito et al., 2007; Prior, Heimovics & Soma, 2013) likely represent the minimal values for the largest follicle in each female in this species. However, they leave two important gaps in our knowledge. Neither study (Perfito et al., 2007; Prior, Heimovics & Soma, 2013) characterized variation in oviduct tissue or tracked variation in follicle size of birds with known breeding status that is, across different phases of active breeding such as egg-laying, incubation, offspring rearing etc. As such, the maximal size of both the oviduct and pre-ovulatory follicles is unclear. As a result, and also due to a lack of clarity over the breeding status of birds, it is not clear how quickly, and to what extent reproductive tissues cycle in this model opportunistic breeder. For example, as illustrated in Fig. 1, in comparison to seasonal breeders (Fig. 1A), females could regress their tissues within and between reproductive bouts to the same (Fig. 1B), or lesser extent (Fig. 1C).

Figure 1 Predicted relative change in reproductive tissue mass.

Change in reproductive tissue mass with breeding events is clearly define in many (A) seasonal breeders with some development prior to rapid increase before breeding (arrow), followed by dramatic decrease after breeding. It is less clear if multiple breeding opportunistic breeders (B) completely regress or (C) partially regress reproductive tissue. Partial regression could be minimal (hatched line) or more significant both with in and out side of active breeding periods.

Studies have suggested that zebra finches consistently maintain some level of gonad functionality, even within a reproductive cycle. For example, female zebra finches can lay eggs within 5 days of pairing (Haywood, 1993; Williams, 1996). Additionally, studies show that regression of the oviduct commences as females are laying their last egg (Williams & Ames, 2004) and that pairs frequently initiate a new clutch before the previous fledglings are fully independent (Griffith et al., 2017). However, no study to date has quantified variation in reproductive tract morphology at different stages throughout the reproductive cycle in an opportunistically breeding bird. Further examination of the extent of variation in reproductive morphology in the zebra finch, as a model opportunistically breeding species, will help us to understand the costs and constraints involved in mounting a reproductive attempt in those birds that have highly flexible breeding times. This is of interest in the context of the rapidly changing ecological conditions that such opportunistically breeding species can face, and the ability of a male and female to physiologically coordinate their reproductive activity and investment (Griffith, 2019).

Here, we investigated variation in female reproductive tissue morphology and circulating hormone levels (i.e., estradiol) across breeding stages of a single reproductive episode in the zebra finch. Given the ability of zebra finches to rapidly initiate reproduction, we predicted that females would maintain some level of gonadal development across a reproductive cycle. Likewise, we predicted that females would maintain moderate levels of estradiol across most of the reproductive cycle, even when their reproductive development is significantly reduced (Prior, Heimovics & Soma, 2013).

Materials and Methods

Animals

Thirty-six pairs of domestic zebra finch were force paired (i.e., we chose partners) into individual cages and allowed to lay at least one clutch of eggs together prior to the commencement of the study. All individuals were bred and maintained at Macquarie University (Sydney, Australia) and were between 18 and 24 months of age. Between February and May 2015, pairs were moved into one of 12 outdoor aviaries (L × W × H: 0.95 × 1.9 × 1.8 m, 1 pair/aviary) that were physically, but not visually or vocally separated from other pairs. All procedures were conducted according to relevant national and international guidelines and were approved by the Macquarie University Animal Ethics Committee (Animal Research Authority 2013/29).

Experimental design

Pairs were then randomly assigned to one of six sampling time points (n = 6 per time point) across the breeding cycle: pre-breeding (paired, but without a nest for a total of 4 weeks including first 2 weeks considered acclimation), nesting (during nest building, once at least the bottom of the nestbox was lined with material), laying (day of laying third egg), incubation, post-hatch (6 days post hatching), and fledging (day after first fledgling was observed out of nest). We chose not to sample on day one of egg laying or immediately post-hatch in order to ensure pairs were committed to the breeding stage (i.e., laying a full clutch and rearing nestlings). All birds were given 2 weeks to acclimate to the new aviary conditions, after which they were provided with nest boxes and nesting material (with exception of those birds in the pre-breeding group). All pairs were checked daily during acclimation and experimental time period to note change in breeding stage, and ensure no nest building or egg laying was occurring in acclimating or pre-breeding pairs.

Blood and tissue collection

At the designated sampling time point, a whole blood sample was collected from each female within 5 min of initial disturbance, and the sample held on ice (for a maximum period of 1 h) before plasma was separated from red blood cells and plasma stored at −80 °C until later analysis of estradiol. Females were euthanised using deep anesthesia with isoflurane. The oviduct was then dissected out (cut off at cloacal juncture) and weighed (after lightly blotting to remove any blood, and if present, eggs were removed prior to weighing). Next, the ovary was dissected out and photographed from two angles, which were then used to later assess the total number of pre-ovulatory follicles present. Finally, on the fresh ovary, the size of the largest pre-ovulatory follicle was measured (three perpendicular measures) using digital calipers (to nearest 0.01 mm). Follicular volumes were calculated using the formula for an ellipsoid, V = 4/3πabc, where the axes a, b, and c are equal to half of the value recorded for each of the three perpendicular measures, to account for deviations from a spheroid in large pre-ovulatory follicles. All nestlings and fledglings were successfully fostered to other pairs of zebra finch in the Macquarie University breeding colony.

Plasma estradiol analysis

Plasma estradiol levels were quantified using a 17-β estradiol high-sensitivity Enzyme Immunoassay (EIA) kit (Cat No. ADI 901-174, Enzo Life Sciences, Farmingdale, NY, USA) following the standard protocol. Briefly, plasma samples (10–30 µl) were diluted in assay buffer to a final volume of 200 µl and, following incubation and washing steps, sample absorbance measured at 405 nm using a Varioskan LUX microplate reader (Thermo Scientific, Waltham, MA, USA). All samples were measured in triplicate, and estradiol levels extrapolated from a ten-point standard curve ranging from 1.95 to 3,000 pg/ml, with a minimum detection limit of 0.8 pg/ml. The average intra-plate coefficient of variation was 9.99% and the inter-plate variation 6.55%. Samples were randomly distributed across plates. One pre-breeding sample was not run as it appeared dehydrated.

Statistical analyses

All analysis was done with R v. 3.5.3 (R Development Core Team, 2019) in the R package lme4 (Bates et al., 2015). We used linear models to examine the effect of breeding stage on oviduct mass (log transformed) and volume of the largest pre-ovulatory follicle. Next, we examined the impact of breeding stage on the number of pre-ovulatory follicles using a Bayesian generalized linear model via Stan with negative binomial family function (to control for zero inflation) and breeding stage as a fixed effect using the R package rstanarm (Goodrich et al., 2018). In these models, we included scaled mass index (SMI: using tarsus length and body mass (Peig & Green, 2009) as a covariate, but when considered alone there was no significant difference in SMI between breeding stages (ANOVA; F5,30 = 0.84, P = 0.53). We did not run a formal statistical analysis on estradiol data due to the low number of detectable samples (see below). Figures were constructed using the R package ggplot2 (Wickham, 2016), and modeling assumptions (normality and heterogeneity of variance of residuals) were assessed visually (following Zuur et al., 2009)). All tests were two-tailed and considered significant at α < 0.05, with significance estimates via lmerTest (Kuznetsova, Brockhoff & Christensen, 2016) for models, and emmeans (Russell, 2019) for pairwise comparison of time points corrected for multiple testing using Tukey HSD P-value adjustment. Data presented are mean ± standard deviation unless otherwise noted.

Results

Change in reproductive tissues

Oviduct mass declined significantly between laying (0.43 ± 0.09 g) and incubation (0.06 ± 0.03 g), reaching its lowest mass during the post-hatch period (0.03 ± 0.01 g) and remaining low for the remainder of the breeding cycle (Table 1; Fig. 2A). Similarly, following a peak in size during nesting (74.9 ± 75.5 mm3), follicle volume decreased significantly by incubation (2.48 ± 0.8 mm3) before reaching its lowest volume during the post-hatch (2.45 ± 1.0 mm3) period (Table 1; Fig. 2B). As expected, the number of ovarian pre-ovulatory follicles was highest during the pre-breeding and nesting stage. The number of follicles then declined during laying, before reaching zero during both the incubation and post-hatch period. The number of pre-ovulatory follicles increased again during the fledgling period, albeit non-significantly and to a lesser degree than the number observed in the pre-breeding stage (Table 2; Fig. 2C).

Table 2 Variation the number of pre-ovulatory follicles across breeding stages in the zebra finch.

Shown are mean ± SD and Bayesian 95% credible intervals (CrI). Letters denote significance differences between given time points, columns with the same letter are not significantly different from one another, columns with different letters are significantly different as CrI boundaries do not overlap.

		CrI boundaries		
Time point	Mean ± SD	2.5%	97.5%		
Pre-breeding	4.5 ± 1.05	2.54	7.92	a	
Nesting	5 ± 0.63	2.77	8.84	a	
Laying	2 ± 1.09	0.94	3.81	ac	
Incubation	0 ± 0	0.001	0.42	b	
Post-hatch	0 ± 0	0.002	0.39	b	
Fledging	0.67 ± 1.20	0.19	1.67	bc	

Figure 2 Quantified variation in zebra finch reproductive tract tissue across and within the six stages of the reproductive cycle.

Variation in (A) oviduct mass (g), (B) volume of the largest pre-ovulatory follicle (mm3), and (C) the number of pre-ovulatory follicles at six reproductive stages: pre-breeding, nesting, laying (day of third egg), incubation, post-hatch, and fledging (day after first fledgling was out of nest). Grey circles represent raw data points, black circle denotes mean with standard error lines. Stages with identical letters do not differ at p ≤ 0.05

Estradiol detection and levels

Estradiol levels were non-detectable in 22 of the 35 samples assayed (62.9%), although one detectable plasma sample appeared dehydrated and was therefore excluded. Of the remaining detectable samples (n = 13), there was a tendency for these samples to be present in the early stages of the breeding cycle and the highest average value was observed during the nesting stage (Fig. 3).

Figure 3 Variation in and detection of plasma estradiol across the reproductive cycle.

Stages in the reproductive cycle: pre-breeding, nesting, laying (day of third egg), incubation, post-hatch, and fledging (day after first fledgling was out of nest). Left axis: levels of circulating estradiol (pg/ml) in plasma. Black circle denotes mean with standard error lines, gray circles represent raw data points. Right axis: sample detectability. Bar denotes number of samples analyzed at each sampling event, dark gray shading denotes samples with detectable levels of estradiol, open bar denotes non-detectable sample.

Discussion

Across a single reproductive episode, from pre-breeding through to fledging of nestlings, female zebra finches showed dramatic variation in the size and development of their reproductive tract tissue (Fig. 4). Oviduct and ovary weight were highest during nesting and laying and regressed during the later stages of the breeding cycle. These results are generally consistent with other studies in this species (Williams & Ames, 2004) and a range of northern hemisphere, temperate zone birds (Dawson, 2008; Hurley et al., 2008; Jacobs & Wingfield, 2000; Ramenofsky, 2011). Similar to the variation in ovary development that we described here, the length of the sperm storage tubules in females vary across a reproductive cycle (Pellatt, 1998) suggesting that the entire female reproductive changes in opportunistically breeding birds. We found that pre-ovulatory follicular development followed a similar pattern, though some females already possessed pre-ovulatory follicles when their first chicks fledged. Importantly, while the observed regression of the female reproductive tract in zebra finches is reminiscent of the regression of these tissues in temperate zone birds, the magnitude of these changes appears to be considerably lower in the zebra finch (Table S1). As such, the reproductive tract regression observed in zebra finches can be considered more similar to the resting phase (e.g., the period between first phase differentiation and breeding when development can be suspended: Sossinka, 1980) of a photosensitive seasonally breeding bird than to that of the fully regressed photorefractory state of such a species.

Figure 4 Variation in zebra finch female reproductive tract across and within the six stages of the reproductive cycle.

Images from across reproductive cycle (A–H) illustrating stage-specific changes in oviduct size, ovary size, pre-ovulatory follicle size, and the number of pre-ovulatory follicles. Left to right: ovary, oviduct orientated infundibulum to cloacal chamber. (A) Pre-breeding, (B) nesting, (C) egg laying (variant 1), female with pre-ovulatory follicle present on ovary suggesting additional eggs to be laid in the clutch, (D) egg laying (variant 2), female with no obvious pre-ovulatory follicles, suggesting no additional eggs to be laid, but an egg present in the uterus (E) mid-incubation, (F) 6 days post hatch, (G) fledgling (variant 1), female with pre-ovulatory follicle present on ovary suggesting rapid initiation to next breeding episode, (H) fledgling (variant 2), female without any observable pre-ovulatory follicles.

It is widely held that the circannual recrudescence and regression cycle observed in seasonally breeding species mediates the energetic and physiological costs of egg production and the maintenance of reproductive tract tissues. These costs include protein depletion and compromised flight ability, as well as non-resource-based costs such as the negative pleiotropic effects of maintaining high hormone levels, for example, immunosuppression (reviewed in Williams (2005)). The idea that such costs influence reproduction is further supported by the fact that seasonally breeding birds typically only engage in one energetically expensive life history stage (e.g., reproduction, molt, migration) at a time (Follett, 2015; Wingfield, 2008). Furthermore, a number of seasonally breeding species have been shown to shift their metabolic rates with reproductive state (e.g., great tit, Parus major (Nilsson & Råberg, 2001); European starlings, Sturnus vulgaris: (Vézina & Williams, 2002)), with this change in metabolism being related to changing energetic demands of reproductive tissue, as well as the liver and gizzard during chick-rearing (Vézina & Williams, 2003). Given this, the semi-regressed “resting phase” state we observed in female zebra finches would likely allow females to rapidly recommence breeding once their current brood is independent, whilst also allowing them to mitigate energetic costs of reproduction via the rerouting of energy from reproductive tissue investment to other activities such as chick rearing. Such rapid turnaround in breeding attempts is supported by our finding that some females exhibited pre-ovulatory follicles as early as the day after the first chick fledged, and is consistent with breeding observations in captive populations (Griffith et al., 2017).

The variation between individuals with respect to the presence of pre-ovulatory follicles during the fledgling period could be considered surprising given that all birds were housed under identical conditions. However, zebra finch pairs frequently exhibit considerable variation in the number of days taken to initiate a new clutch, which would likely reflect underlying variation in the presence of pre-ovulatory follicles. In turn, this variation can be attributed to variation in breeding experience as a pair (Hurley, Rowe & Griffith, 2020) and variation in individual condition and life-history trade-offs (Williams, 1996, 2005). The observed variation in follicle development is broadly consistent with findings in wild zebra finches in unpredictable, arid environments. Specifically, Perfito et al. (2007) reported a proportion of female birds were not completely regressed and had small follicles present when sampled despite an apparent absence of breeding within the population (49% in predictable temperate population during winter and 58% during a drought in late Spring in an unpredictable arid population).

Estradiol levels observed in this study were highly variable and a large proportion of the samples were below the detection level. Whilst it was disappointing that we were unable to assay the level of estradiol in so many of our samples, we feel that the data are worth publishing, and that, for the reasons discussed below, they are likely to reflect the underlying biology, and really low values, rather than methodological problems. Past studies have also reported large numbers of undetectable samples. For example, in a study of wild zebra finches using identical methods, estradiol was reported as non-detectable in 38.3% of samples (Crino et al., 2018), and in an earlier study of another passerine, the Western scrub-jay (Aphelocoma californica) the authors were unable to detect estradiol levels in the majority of their samples using similar methods (Rensel et al., 2015). The higher percentage of detectable samples in Crino et al. (2018) are likely explained by the use of gonadotropin-releasing hormone challenges to induce maximum estradiol release, whereas as our study examines natural endogenous levels. Finally, the peak in estradiol levels observed during the early stages of the reproductive episode, that is, nesting and laying, is in line with the essential role estradiol plays in the growth and development of the oviduct (Pollock & Orosz, 2002; Williams, 1999), as well as egg yolk and eggshell formation (Mishra, Sah & Wasti, 2019). This suggests that levels in the zebra finch are generally low, and in many individuals below the threshold of detectability, but that they do show the anticipated pattern of variation across the cycle.

Somewhat surprisingly, we found that ovary and oviduct development of females in the pre-breeding stage did not differ from those in the nesting stage. There are at least two plausible explanations for these results. First, domestic zebra finches are highly motivated to breed, and, when paired, females will frequently lay eggs on any suitable surface despite a lack of nest boxes and nesting material. Moreover, as oviduct and follicular development is sensitive to a change in circulating hormones, for example, estrogen, lutenizing hormone, follicle stimulating hormone (Johnson & Woods, 2007; Pollock & Orosz, 2002), the still somewhat favorable conditions experienced by these birds (e.g., ad lib water and food) may have been sufficient to support elevated levels of hormones linked to reproductive tissue development. Thus it seems likely that females defined as pre-breeding in our study are functionally equivalent to seasonal breeders in a pre-laying state (when predictive cues have primed reproductive tissue), and not a completely non-breeding state. It is unclear how long females will maintain readiness, especially in variable ecological condtions. In normal captive conditions, it can be very difficult to prevent birds from breeding due to the provision of relatively good resources (Griffith et al., 2017). Even though the females in our pre-breeding treatment were unable to physically breed due to the lack of a nest site, this is quite an unnatural constraint, that would not be encountered in the wild and it seems likely that physiologically these females were ready to breed. Second, we defined pairs as nesting once males were observed actively building nests. Thus, our distinction between pre-breeding and nesting stage may have been too coarse to detect differences between these stages. Pairs may have already established themselves into reproductive mode in the days before nest building started through the many subtle behaviors and forms of communication that we expect in such closely coordinated partners (Griffith, 2019). It is therefore very difficult in captivity to get zebra finches to a state where they can be confidently established to be in a stable and lengthy non-breeding mode. It therefore remains to be checked in a wild population whether the gonadal and endocrine state that we have reported in the pre-breeding birds is truly reflective of non-breeding birds. To an extent however, we are reassured that the wild birds assessed by Perfito et al. (2007) as being in a non-breeding state had a range of folicle size that was similar to those found in our pre-breeding birds (Table S1).

Conclusion

Detailed examination of the female reproductive tract of zebra finch across a single breeding episode revealed considerable variation in oviduct and ovary development, and a pattern of limited reproductive tract regression during the later stages of the reproductive cycle (i.e., incubation, post-hatch). We suggest the pattern of regression we observed is consistent with the lack of a photorefractory period with less regression than seen in another opportunistically breeding species (Hahn, 1998), and stands in stark contrast to observations in seasonally breeding birds. By dampening the degree of regression of the reproductive tract, female zebra finches might maintain a general state of reproductive readiness, whilst also mediating some of the energetic costs associated with reproduction. Future studies could test this hypothesis by examining the metabolic costs associated with partial reproductive regression in opportunistically breeding birds. Furthermore, future studies aimed at determining whether similar patterns of variation in female reproductive tract morphology are observed in the many other species that show a similar or greater extent of reproductive plasticity with respect to phenology (Englert Duursma, Gallagher & Griffith, 2017) would be valuable. To date much of the focus on the extent of female reproductive tract variance has focused on seasonal breeders of the northern hemisphere, and it is likely that quite different patterns will be seen in Australia and other ecologically unpredictable and aseasonal environments (Astheimer & Buttemer, 2002).

Supplemental Information

Supplemental Information 1 Data used in analysis.

Click here for additional data file.

Supplemental Information 2 R code used for statistical analysis.

Click here for additional data file.

Supplemental Information 3 Variation in gonadal development in female passerine birds.

Change in reproductive tissue in seasonal breeders and change in Zebra finch from this and previous studies (below double hatched line). Note only studies reporting actual quantitatively measured oviduct mass and/or follicle volume, are used to make comparison to the current study. Other studies do exist that use qualitative scoring of development/condition.

Click here for additional data file.

We would like to thank: Drew Allen for help with the pre-ovulatory follicle data analysis; Mitch Francis for assistance in monitoring pairs, as well as photography and labeling during sample collection; Katherine L. Buchanan for comments on an earlier draft of the manuscript; Xavier Harrison, Nicola Hemmings and one anonymous reviewer for useful comments on the manuscript; and the staff of Macquarie Animal Research Services for animal care.

Additional Information and Declarations

Competing Interests

Author Contributions

Animal Ethics

Data Availability

Simon C. Griffith is a PeerJ Academic Editor. The authors declare that they have no competing interests.

Laura L. Hurley conceived and designed the experiments, performed the experiments, analyzed the data, prepared figures and/or tables, authored or reviewed drafts of the paper, and approved the final draft.

Ondi L. Crino performed the experiments, authored or reviewed drafts of the paper, and approved the final draft.

Melissah Rowe conceived and designed the experiments, prepared figures and/or tables, authored or reviewed drafts of the paper, and approved the final draft.

Simon C. Griffith conceived and designed the experiments, authored or reviewed drafts of the paper, and approved the final draft.

The following information was supplied relating to ethical approvals (i.e., approving body and any reference numbers):

All procedures were conducted according to relevant national and international guidelines and were approved by the Macquarie University Animal Ethics Committee (Animal Research Authority 2013/29).

The following information was supplied regarding data availability:

The dataset and related R-code is available at the Open Science Framework:

Hurley, L., Crino, O. L., Rowe, M., & Griffith, S. C. (Accepted 2020, Sept 25). Variation in female reproductive tract morphology across the reproductive cycle in the zebra finch. PeerJ. https://osf.io/43qsd/.

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
