# Peer review of "Variation in female reproductive tract morphology across the reproductive cycle in the zebra finch"

_PeerJ, doi:10.7717/peerj.10195_

## Round 0.1 · original submission · Major Revisions

Your manuscript has now been assessed by two expert reviewers, whose detailed comments are appended below. Both found your study interesting and agree it addresses an important question, but several issues require clarification in the revision.

In particular I'd like to see some justification of the apparent disparity between the estradiol results and their relatively strong prominence in the title and abstract, as raised by rev. 1.

For Fig. 3, I commend the authors for providing raw data alongside summary statistics, but I'd argue the triple combo of quasi-boxplot, violin plot, and raw data points is a lot for the reader to take in visually. I'd be tempted to ditch the box plots, though a thin bar for the mean may still add value.

I look forward to seeing a revision.

·

Basic reporting

There are a few minor editorial changes required, as well as some more methodological detail. The title and abstract do not provide a totally accurate reflection of the results so require re-writing. There is too much focus on the estradiol part of the study (which ultimately didn't work) in the abstract and title.

The introduction would benefit with some more explanation of and justification for the estradiol question (and why other key reproductive hormones were not considered).

Table 1 could arguably be a supplementary file/appendix. It also needs some more information to be useful in the context it is intended (specifically, information on when sampling was carried out in each study is important for comparison with the current study). Figure 3 requires a clearer legend and there are some conceptual issues with Figure 1.

I'm not convinced the data obtained allow the hypotheses to be satisfactorily tested.

These points are all expanded upon in my comments to authors.

Experimental design

Further data are required to fully answer the questions posed by this study. If this is not possible, the study's aims/scope need to be adjusted and interpretation of the results tempered.

More detail is required in parts of the methods.

These points are all expanded upon in my comments to authors.

Validity of the findings

Further data are required to fully answer the questions posed by this study. If this is not possible, the study's aims/scope need to be adjusted and interpretation of the results tempered.

These points are all expanded upon in my comments to authors.

Additional comments

This manuscript presents the interesting question of how the oviduct changes throughout a reproductive cycle in an opportunistically breeding bird, the zebra finch. Our current understanding of how the reproductive tissues of birds changes from non-breeding to breeding state is largely limited to seasonal breeders such as temperate songbirds. In seasonal breeders, the oviduct in females (and testes in males) grows at the start of the breeding season in response to hormonal changes triggered by increasing day length, and regresses significantly after breeding, presumably as a cost-saving mechanism. This growth and regression take place over a relatively predictable and stable timeframe, allowing the timing of reproduction to be matched with environmental conditions. Opportunistic breeders, however, tend to breed in unpredictable environments, responding quickly to optimal breeding conditions. How they achieve this is poorly understood, despite the fact that in captivity, opportunistically breeding species like the zebra finch have been recorded to produce eggs less than a week after pairing with a male. How this rapid response happens at the mechanistic level, and what the costs might be, are fascinating questions. I was pleased to see a study focused on them.

The manuscript is generally well written and straightforward, but I have some concerns about the methods and interpretation/presentation of the results. I also have some additional questions/ideas that I would like the authors to consider. The key issues are listed in order of importance here, but see also the specific comments afterwards for more minor comments and suggestions.

Key concerns

1. The study doesn't explicitly measure 'non-breeding' state, so we do not know what the baseline values are for this species. The earliest sampling period is 'pre-breeding', which is before a nest site and material was provided. However, if I understand the methods correctly, females were already paired with a male that they had produced a clutch with previously, before they were moved to the experimental cage. It is therefore not clear that these females were in a 'baseline' reproductive condition (especially since some females of this species will lay on the cage floor if nesting sites are not available). The authors note this at the end of the discussion, but do not sufficiently deal with the implications of this in terms of their comparisons with seasonal breeders and their interpretation of patterns in Figure 1 (see specific comments below). It would be useful to have some measure of oviduct size from a group of females that haven't been paired, so that a true baseline state (and variation within) can be inferred.

2. The study explicitly aims to quantify how oviduct size changes throughout a reproductive cycle in an opportunistic breeder, for comparison with what is known about seasonal breeders. In this study, however, opportunistic breeding is confounded with domestication and captivity. It seems likely that the costs of maintaining reproductive tissue would be lower for captive birds with ad lib food etc, than it would be for wild birds. Moreover, domesticated strains may have been selected for high reproductive output and therefore more likely to be 'productive' (e.g. domestic chickens are an extreme example of this). These factors will surely influence oviduct condition (just as they do body condition), so it would have been ideal to have some wild comparison data (even just at one non-breeding and one breeding time point) to see if the patterns in captive birds are generally representative of wild birds. If it's not possible to add this, the possible effects of captivity/domestication should at least be discussed in the manuscript.

3. The estradiol part of the study largely failed. Because of this, the data obtained for this part of the study are very few and potentially not reliable. Yet, the estradiol 'result' is made to be a focal point of the manuscript via its inclusion in the title and coverage in the abstract (without any indication of the test failure). The emphasis needs to be taken off of the estradiol part of the study in the title and abstract. I support including information about the attempted trial and failures (as well as ideally some attempt to explain why), because it is useful for other researchers to be aware of potential methodological issues in the future. However, I do not think very much at all should be inferred from the data obtained. More generally, I also wonder why estradiol was focused on exclusively (especially given the poor performance of this test), in preference to other important reproductive hormones such as FSH or LH?

In my opinion, a revision of this manuscript should either (a) be rewritten to focus primarily on the oviduct size data alone, framed more simply and with the interpretation of the results somewhat tempered (i.e. taking into consideration the lack of a true 'baseline' measure and the confounding issue of captivity/domestication) OR (ideally) (b) include additional data to resolve these issues. While option (b) is obviously more time consuming and dependent on the availability of funds for further work, it would ultimately make this paper far more useful and interesting.

Below are some more specific comments on the manuscript (line numbers referred to), including some developments on the key points made above. I hope the authors find these comments useful when revising their manuscript.

Abstract

line 20 (and throughout MS): use of "recrudescence". This is a somewhat obscure word that I've never heard used in this context. When I looked it up, it seems to be most commonly used to describe disease relapse. I would personally prefer a more straightforward word, perhaps simply "growth" or "re-growth" (as is typically used in similar studies).

line 27: ‘Physiological readiness’ rather than ‘physiologically readiness’

line 33-35: the estradiol part of this study didn't really work and we can't infer much from the data obtained. But in the abstract it is presented as a straightforward result. I would remove this from the abstract (and remove the reference to estradiol from the title).

line 35-37: I think the abstract should include something about the regression being less dramatic in the zebra finch than has been previously found for seasonal breeders. To me, this is probably the most interesting and illuminating result (as you nicely explain at the end of your discussion). The caveat to this is, of course, if this is truly a consequence of being an opportunistic breeder, rather than just a consequence of captivity/domestication.

Introduction

line 69-70: "average" and "mean" are used interchangeably here, which is a bit confusing and unclear. Presumably "average" means "mean"? If so, use "mean" consistently

line 72: remove the bracket around (Perfito et al. 2007) to read Perfito et al. (2007) since the study in question is being directly referred to.

line 76: it is not clear what "maintained partially developed gonads when not breeding" means – partially developed compared to what? If all non-reproductive females have gonads that appear to be at the same level of development, presumably this is their baseline (i.e. undeveloped) state? Perhaps opportunistic breeders baseline is simply more developed than seasonal breeders (or to flip it round, seasonal breeders regress more dramatically than opportunistic breeders)? This also has implications for how the situation is depicted in Figure 1C (see later comment on this).

line 76: remove the bracket around (Prior et al. 2013) to read Prior et al. (2013)

line 80-81: (linked to previous comment) could there be something else about seasonal breeders (i.e. the environments they are typically found) that makes them need to regress more dramatically than opportunistic breeders would? I.e. it's not so much that opportunistic breeders are 'hanging on' to their reproductive capacity, but rather that seasonal breeders have to substantially pare back in order to survive.

line 101-103: the way this is written makes it sound like individuals will be followed through the reproductive cycle, which is slightly misleading. Could be reworded e.g. "no study to date has quantified variation in reproductive tract morphology at different stages throughout the reproductive cycle in an opportunistically breeding bird."

line 111: the estradiol question feels a bit tagged on at the end of the introduction. If this is retained this as a focal part of the study (which it perhaps shouldn't be; see general comments), the introduction would benefit from a little bit more coverage of the role of estradiol earlier on e.g. when introducing about how tract growth and regression typically occur in seasonal breeders, the role of estradiol could be mentioned, and then as the manuscript goes on to introduce opportunistic breeders, it could highlight the fact that we don't really understand how estradiol works in this type of reproductive mode (which then sets up the question/hypothesis about estradiol and better integrates it into the overall biological background). Also, why just estradiol and not other important reproductive hormones such as FSH and LH?

Methods

line 121: "thirty-sex" = "thirty-six"?

line 121: "force paired" meaning may not be clear to someone with no experience of bird breeding. Suggest replacing with "were established in individual cages to prevent pairing or copulations with other birds" (or similar)

line 121-122 "at least one clutch" – how much variation in prior breeding/laying experience was there between females? Might we expect oviduct size/flexibility to change with age and/or reproductive experience? I also wonder if there could be some sort of anticipatory effect here - if you have produced 3 clutches in a row are the perceived odds that you will be able to lay another higher? Could this influence retention of reproductive tissue?

line 132 when is "pre-breeding"? (precisely)

line 132 how is "nesting" defined? (precisely) - is this nest-building? Also more generally, it would be helpful to use an easily differentiated word for either nesting or nestling to avoid confusion.

line 138: how often to females lay on the floor in this population (both non-breeders and those in pairs)? To produce an egg, females must have a functioning (i.e. developed) oviduct, and it seems this is quite common in captive populations of zebra finches even without the typical cues that induce breeding condition (i.e. pairing up with a male). Could this potential variation influence the interpretation of your results?

line 145-146: were the oviducts stripped of connective tissue and blotted first, before weighing? Can you provide a bit more explanation about how follicle counts were done from photos? The ovary is three-dimensional so some follicles would be hidden in a two-dimensional photo (evident from Figure 2)? Wouldn't it have been easier to preserve these and count directly?

line 168: why mixed models? What was the random effect? Each female provides just one data point, right?

Results

line 186-187: this first sentence is redundant, suggest omitting

line 193-194: isn't it obvious that follicle number and ovary mass will decline during this period? Not because of ovary 'regression', but simply because ova are being released? During and after laying follicles are releasing or have released their ova, respectively... not sure why these changes in the ovaries are particularly biologically interesting. The oviduct mass data seems more useful.

line 200: do you have any idea what went wrong with the estradiol trials? Was it more likely a technical problem (if so, details of this might be important from a methodological perspective for other researchers) or could it be that the birds actually have very low estradiol levels? If the latter, could opportunistic breeders be fundamentally different to seasonal breeders in terms of their reproductive hormonal profiles? I.e. different proximate mechanisms for triggering oviduct growth and ovulation? Is anything known about this?

Discussion

line 211-212: over what timeframe do the changes in oviduct size occur in seasonal breeders previously studied? Does it take a much longer time? To me, this seems to be the interesting biological question - can opportunistic breeders do this far more quickly than seasonal breeders, and if so, how?

line 217: it would be interesting to know if between-female variation depends on other maternal factors e.g. age/experience (see my similar earlier comment)

line 246-248: this is a good point (and links to the above comment) – was there any variation like this within the study cohort e.g. previous breeding experience? E.g. as mentioned previously, the methods specifically state that birds were allowed to lay "at least one" clutch (implying that some laid more). Is there any variation in this respect that could be informative?

line 254: how repeatable and reliable are these tests? Presumably not very, given the high failure rate. Are the detectable measures definitely reliable?

line 275: it would have been interesting to sample a random set of females before or very soon (i.e. one/two days) after pairing, to get an idea of what their baseline state is and therefore what the rate of oviduct regrowth is. I wonder if, under captive conditions, females are able to maintain their oviduct tissue more readily than a wild bird would be able to because living conditions are 'easy'? In this context, how representative are domesticated captive zebra finches likely to be of their wild counterparts? It may simply be easier to maintain oviduct tissue in captivity, so the effect we see here is actually a captivity effect, not an opportunistic breeder effect. Ideally, data from truly non-breeding females (to get real baseline data) and/or some comparison data from wild birds (even just non-breeding vs breeding time-points to get a rough idea of how comparable captive birds are to wild birds) would be added here to complete the puzzle. Currently, the existing data only really tell us is how the oviduct changes from after pairing (by which time changes may have already happened) through to chick-rearing, in a domesticated captive bird with (probably) limited costs associated with maintaining reproductive function outside of breeding.

line 277: definitions of what "pre-breeding" and "nesting" phases actually are need to be included in the methods, not just at the end of the discussion

line 278: change ‘destinction’ to ‘distinction’

Fig 1C - this figure may be misleading. The data collected in this study don't allow us to infer reproductive tissue state outside of the breeding period (i.e. completely before and after the breeding cycle). This figure makes it look like the opportunistic breeder baseline is the same as it is in seasonal breeders outside the breeding season. However, it is possible that it never regresses to the same extent.

Figure 3: figure legend could be clarified by changing ‘quantified variation in zebra finch female reproductive tract across and within…’ to ‘quantified variation in zebra finch reproductive tract tissue across and within…’

Table 1: while it's interesting to see these data and what is already known, it is not clear at what point in the reproductive cycle these traits were measured in other studies, which makes it difficult to draw comparisons with the current study. I also think this table probably isn't needed in the main text and should instead be in supplementary material.

Reviewer 2 ·

Basic reporting

The manuscript is well written and easy to follow with appropriate figures. Data are shared.

Overall, the the literature is well referenced. However, I do have a couple of suggestions for improvement in this regard.

In general, I think the characterization of our current understanding of opportunistic/flexible breeders is a bit of an oversimplification. For example, Ln 66 - I would disagree with the interpretation of the Hahn 1998 data as shown "little annual variation". Fig 2 shows females with an ovary score of 2 in late fall, which the Methods indicates reflects an absence of apparent follicles.

Related to the comment above, it isn't clear how data were selected for inclusion in this Table 1, as it does not appear to be comprehensive. For example, the Hahn data mentioned above are not included and it appears that only some species in the Wikelski et al. 2003 study are included.

Seasonal breeding is oversimplified in places as well. In particular, ln 48 - I suggest deleting "much" from the statement, ln 52 - "inflexible" is an overstatement as fine-tuning certainly occurs.

Minor comments

Provide full references for studies in Table 1. I couldn't locate these.

Experimental design

Overall methods experimental design and methods are well defined and meaningful. However, further detail is needed in a few places.

1) More information is needed on how the "pre-breeding" period was defined and the state of those birds.. The authors note some difficulties with this category in the Discussion (and provide necessary caveats for interpreting the data), but it would be useful for readers to have more information about this categorization in the Methods. Is this defined purely on the fact that they did not have a nest-box and nesting materials? Was this period immediately following the 2 week acclimation? How long had it been since these birds had produced their last clutch?

2) It is stated that mixed models are used for statistical analysis, but it isn't clear what is the random term(s) in the models

3) No information is provided on priors used for Bayesian modeling. Were additional diagnostics performed for this modeling?

Minor comments
ln 144 - is "deep anesthesia" really what is meant here given that birds are euthanized?

Validity of the findings

Overall results and conclusions are well supported. But see comments below.

1) Ln 191 - the statement about a significant decrease between laying and incubation does not match statistical results presented in Table 2.

2) ln 286 - the conclusion that this reflects a lack of photo refractoriness is a bit of a leap. Perhaps this could be better supported with use of literature?

3) ln 288 - reword to make clear this is a hypothesis (e.g., change "can" to "might")

Minor comments

Figure 4 legend use of the term "bar" to describe both the black line showing mean and the boxes for counts is confusing.

Figure 3. Is it possible to include letter labeling to indicate significant pairwise differences (from Table 2)?

Additional comments

Other minor suggestions

ln 22 - "seasonal" and "environmental" cues are not necessarily alternatives. I suggest rewording.

ln 48 - delete "much"
ln 221 - define what is meant by the resting phase.

ln 285 - should "nesting" be "nestling"?

---

## Round 0.2 · accepted · Accept

Many thanks for making the requested changes to the manuscript in line with detailed comments arising from the previous round of review.

I have now reassessed your manuscript, and am satisfied with the tempering of discussion and interpretation of results to reflect the limited data available from the estradiol assays, as well as numerous other changes throughout. I am happy to recommend it for publication.